# Distribution and Public Health Significance of *Vibrio* Pathogens Recovered from Selected Treated Effluents in the Eastern Cape Province, South Africa

Ayodeji C. Osunla [1,2,3,*], Oluwatayo E. Abioye [1,2,4] and Anthony I. Okoh [1,2,5]

1   SAMRC Microbial Water Quality Monitoring Centre, University of Fort Hare, Alice 5700, South Africa; abioyethayor@gmail.com (O.E.A.); AOkoh@ufh.ac.za (A.I.O.)
2   Applied and Environmental Microbiology Research Group, Department of Biochemistry and Microbiology, University of Fort Hare, Alice 5700, South Africa
3   Department of Microbiology, Adekunle Ajasin University, Akungba-Akoko, Ondo-State P. M. B 001, Nigeria
4   Department of Microbiology, Obafemi Awolowo University, Ile-Ife P. M. B. 13, Nigeria
5   Department of Environmental Health Sciences, University of Sharjah, Sharjah P. O. Box 555588, United Arab Emirates
*   Correspondence: osunlacharles@gmail.com

**Abstract:** Treated sewage harbours pathogenic microbes, such as enteric bacteria and protozoa, are capable of causing several diseases. Some of these are emerging pathogens sometimes recovered in the absence of common water quality indicator organisms. The possibility of selected treatments plants serving as momentary reservoirs of *Vibrio* pathogens during a non-outbreak period was assessed. The occurrence and diversity of *Vibrio* pathogens were monitored for one year (December 2016 to November 2017) in the treated effluents and upstream and downstream areas of the receiving water bodies of two wastewater treatment plants (WWTPs), designated AL and TS. Physicochemical parameters of TS and AL WWTPs' water samples were analysed using a multi-parameter meter (Hanna, model HI 9828, Padova, Italy) and a turbidimeter (HACH, model 2100P, Johannesburg, South Africa). Water samples were augmented with alkaline peptone water and cultured on thiosulfate citrate bile salts sucrose agar at 37 °C for 24 h. The recovered probable pathogens were confirmed via PCR amplification, using primers specific for *Vibrio* species of public health significance. The distribution of *Vibrio* species positively and significantly (p < 0.01) correlated with turbidity (r = 0.630), temperature (r = 0.615), dissolved oxygen (r = 0.615), pH (r = 0.607), biological oxygen demand (r = 0.573), total dissolved solid (r = 0.543), total suspended solid (r = 0.511), electrical conductivity (r = 0.499), residual chlorine (r = 0.463) and salinity (r = 0.459). The densities of *Vibrio* species were found to be significantly higher (p < 0.05) in effluents from both AL and TS WWTPs than upstream and downstream of the receiving rivers across the sampling regime. Furthermore, the maximum *Vibrio* species density across the sampling regime were observed during the warmer Summer and Spring season. Moreover, six medically important *Vibrio* species were detected in the water samples, indicating that the methods employed were efficient in revealing that WWTPs are potential reservoirs of *Vibrio* pathogens, which could pose a substantial public health risk if the receiving water is used for domestic purposes. Our findings further strengthen existing calls for the inclusion of emerging bacterial pathogens, including *Vibrio* species, as water quality indicators by the South African Department of Water Affairs. Hence, we recommend regular monitoring of treated effluents and receiving water bodies to ensure early control of potential outbreaks of vibriosis and cholera.

**Keywords:** *Vibrio* species; treated effluents; freshwater; public health; wastewater treatment plants

## 1. Introduction

Treated effluents and their contiguous natural water resources, such as rivers, streams and brackish water are known to harbour copious amounts of waste products, including enteric pathogens [1]. The inherent traits possessed by these microorganisms to persist in conservative treated effluents might pose a major risk to human health, considering the domestic use of highly contaminated surface waters [2]. Wastewater effluents are potential reservoirs and transporters of pathogenic *Vibrio* species, serving as an enabling environment for the pathogens to thrive against unfavourable conditions [3,4]. Despite the incredible efforts of international bodies and water management organizations, such as the World Health Organization (WHO) and Water Research Commission (WRC), to upgrade and maintain water treatment standards, the prevalence of waterborne infections remains evident in developing countries [5]. Furthermore, the occurrence of potentially pathogenic microbes in effluents from wastewater treatment plant and adjacent river water indicates that the wastewater treatment plants (WWTPs) are not effective in eradicating all pathogens as expected [6]. Hence, treatment plants have been implicated in the distribution of enteric pathogens, such as the main 12 known pathogenic *Vibrio* species to human, in contaminating freshwater ecosystem [7,8]. *Vibrio* species are known autochthonous populations found in freshwaters and marine sediments worldwide [9,10]. Several species of *Vibrio* are known to cause a number of diseases in human, including cholera, gastroenteritis and primary septicaemia, and their infections arise from the ingestion of contaminated foods, water, undercooked seafoods and through wounded skin getting direct contact with contaminated water [11–14]. The upsurge of potentially pathogenic *Vibrio* species in freshwater resources is partially attributable to global warming induced by hydro-climatic changes [15]. Over the years WWTPs across South Africa have been found to discharge inadequately treated effluents with various enteric pathogens, such as *Vibrio* species [16]. Moreover, existing treatment regulations and guidelines, which use common indicator organisms to assess portable water and treated effluents, may underestimate potential health risks from many pathogenic microbes [17]. The South African Department of Water and Sanitation, in its Performance Audit on Water Infrastructure meeting in 2017, reported that 40 to 50% of the 1400 wastewater works are in a poor state at medium to high risk. Additionally, the treatment of wastewater generated in rural and peri-urban areas of the Republic of South Africa (rural settlement wastewaters) is either absent or scarce [18]. This is rather disturbing, considering that ~19% of rural dwellers in South Africa absolutely rely on local freshwater resources for most of their domestic activities without further treatment [18]. South Africa is located in a semiarid region and receives very little rainfall, resulting in shortage of potable water. Most individuals, therefore, resort to water from unprotected sources, such as rivers, boreholes and dams, for daily activities, such as irrigation, cooking and even drinking [9]. Besides, shockingly high cases of waterborne disease outbreaks, associated with a lack of fundamental sanitation and infrastructure, have been documented in poverty-wracked settlements [2]. Therefore, there is an immense need for regular monitoring of the occurrence and distribution of clinically important *Vibrio* species in the effluents from WWTPs. Hence, this study evaluated treated effluents from selected WWTPs in the Eastern Cape Province of South Africa and its receiving watersheds as potential reservoirs of *Vibrio* pathogens. Interestingly, this is the first time that this kind of study will be carried out at the selected WWTPs.

## 2. Materials and Methods

### 2.1. Study Area and Water Collections

A description of the two selected WWTPs, designated AL WWTP and TS WWTP, are presented in Table 1. Our sampling of both AL and TS was approved by the appropriate authority. Treated effluents and receiving water samples were collected between December 2016 and November 2017 at the end-phase of treatment in WTTPs and ~500 m up and downstream of the treated effluent release point into freshwater bodies. Samples were

collected in duplicates in 1 L germ-free bottles against the river tidal flow, transported in ice packs cooler to our laboratory and processed within 6 h. The sampling bottles already contained 3% sodium thiosulphate to minimize the influence of lingering chlorine on indigenous microorganisms.

**Table 1.** Detailed description of AL and TS wastewater treatment plants.

| WWTPs | AL | TS |
|---|---|---|
| Municipality | Sarah Baartman District | Chris Hani District |
| Technology | Activated Sludge | Stabilization pond |
| Geographic location | S33.31612626°, E26.107717° | S32.045122, E27.810904 |
| Design capacity (ML/d) | 1.1 | NI |
| Receiving river | Boesman | Tsomo |
| Population statistics (2016) | 138, 182 households | 194, 291 households |

NI = No information.

### 2.2. Determination of Physicochemical Parameters

Synchronously with water sample collection for bacteriological examination at each sampling site, environmental variables were measured using a multi-parameter meter (Hanna, model HI 9828, Padova, Italy) and a turbidimeter ( ). Physicochemical parameters of AL and TS WWTPs water samples such as temperature (°C), pH, electrical conductivity (EC) (μS/cm), dissolved oxygen (DO, mg/L), turbidity (NTU), TDS (mg/l) salinity (PSU), total suspended solids (TSS, mg/L) and total dissolved solids (TDS, mg/L) were measured at the sampling site. The biochemical oxygen demand (BOD, mg/L) of the samples was measured after 5 days of incubation in the dark under ambient condition using a BOD meter (HACH, HQ 40d, Johannesburg, South Africa). All physicochemical parameters were determined following standard instrumental procedures.

### 2.3. Bacterial Strains

Six of the seven *Vibrio* strains used in this study were sourced from the Leibniz Institute DSMZ-German Collection of Microorganisms and Cell Cultures GmbH, including *Vibrio alginolyticus* DSM 2171, *V. parahaemolyticus* (*DSM 10027*), *V. fluvialis* (*DSM 19283*), *V. mimicus* (*DSM 19130*), *V. vulnificus* (*DSM 10143*) and *V. alginolyticus* (*DSM 19130*). A locally isolated strain of *V. cholerae* was used as a reference organism for forming *V. cholerae* isolates in this study.

### 2.4. Bacterial Isolation and Enrichments of Samples

Treated effluent and receiving water body samples were serially diluted using the standard method prescribed by [19]. Ten-fold serial dilution up to power of four was carried out on all samples by adding100 ml of sample to 900 ml sterile distilled water. Then, 100 ml of each dilution was filtered using a standard membrane MF-Millipore filter (47 mm, 0.45 μm pore size), following [20], which was then aseptically transferred to thiosulfate citrate bile salts sucrose (TCBS) agar (Oxoid, Cambridge, UK) and cultured at 37 °C for 24 h to determine the presumptive microbial load. After the incubation period, distinct yellow and green colonies observed on the plates were counted as presumptive *Vibrio* isolates. The yellow and green colonies (X) were expressed in colony-forming units per 100 ml (CFU/100 ml) and log transformed (log (1 + X)). Moreover, for enhanced isolation of *Vibrio* spp. of interest, 10 ml of water sample was seeded into 90 ml alkaline peptone water (APW) in a 250 ml conical flask. The conical flasks were agitated gently for 2 min for even mixture of APW and water sample and incubated at 37 °C for 24 h. Then, a loopful of the biofilm formed as a thin layer at the surface of the APW sample experimental setup was carefully streaked on TCBS agar and incubated at 37 °C for 24 h. Distinct yellow and green colonies observed on the plates were randomly selected, and selected isolates were sub-cultured onto fresh TCBS to obtain pure cultures. The pure probable isolates

were subsequently cultured at 37 °C for 24 h on freshly prepared nutrient agar and then preserved in 20% glycerol stocks at −80 °C for subsequent studies.

### 2.5. Confirmation of Probable Vibrio Isolates

DNA from recovered isolates was carefully extracted via the boiling method [21]. The isolates were then confirmed as member of *Vibrio* genus using the DNA template and the *Vibrio* genus-specific primer that targets the region (700 bp–1325 bp) of 16S rRNA gene in a PCR assay, as described in one of the earlier study from our laboratory. The primer sequence to confirm the identity of the probable *Vibrio* species isolates is presented in Table 2. The 25 µl PCR cocktail included 5 µl DNA templates, 12.5 µl 2× *Taq* Master Mix of Standard Buffer (BioLabs, Hitchin UK), 1 µl each of 10 µM forward and reverse primers and 5.5 µl nuclease-free water. The PCR protocol for single enzyme activation was as follows: pre-denaturation at 93 °C for 15 min, 35 cycles of denaturation at 92 °C for 40 s, annealing at 57 °C for 60 s, extension at 72 °C for 90 s and final extension at 75 °C for 7 min. *V. fluvialis* (DSM 19283) was used as a positive control.

**Table 2.** Characteristics of primers used for PCR amplification of genus- and species-specific genes.

| Specie | Sequence | Size bp | References |
|:---:|:---:|:---:|:---:|
| *Vibrio* genus | F: CGG TGA AAT GCG TAG AGA T<br>R: TTA CTA GCG ATT CCG AGT TC | 663 | [22,23] |
| *V. cholerae* | F: CAC CAA GAA GGT GAC TTT ATT GTG<br>R: GGT TTG TCG AAT TAG CTT CAC C | 304 | [24,25] |
| *V. parahaemolyticus* | F: GCA GCT GAT CAA AAC GTT GAG T<br>R: ATT ATC GAT CGT GCC ACT CAC | 897 | [23,26] |
| *V. vulnificus* | F: GTC TTA AAG CGG TTG CTG C<br>R: CGC TTC AAG TGC TGG TAG AAG | 410 | [23] |
| *V. Fluvialis* | F: GAC CAG GGC TTT GAG GTG GAC GAC<br>R: AGG ATA CGG CAC TTG AGT AAG ACT C | 217 | [23,27] |
| *V. Mimicus* | F: GGT AGC CAT CAG TCT TAT CAC G<br>R: ATC GTG TCC CAA TAC TTC ACC G | 390 | [28] |
| *V. alginolyticus* | F: GAG AAC CCG ACA GAA GCG AAG<br>R: CCT AGT GCG GTG ATC AGT GTT G | 337 | [29] |

### 2.6. Delineation of Vibrio Species Isolates

Species-specific primers were used to delineate the confirmed isolates into six different species. Appropriate primer sets (Table 2) were used in the PCR assay to authenticate the isolates that falls within the *Vibrio* species of interest in this study. A triplex PCR was employed for the concurrent delineation of *V. alginolyticus, V. fluvialis* and *V. vulnificus*, while a duplex PCR procedure was also used for concurrent confirmation of *V. mimicus*, and *V. cholerae Vibrio parahaemolyticus* was confirmed using simplex PCR. All protocols were as described in earlier studies [30,31]. The thermal condition for the PCR is as described under Confirmation of Probable *Vibrio* isolates section above except that annealing temperatures for triplex, duplex and singleplex PCR were 66.3, 54.5 and 64 °C, respectively.

The positive controls used were *V. parahaemolyticus* (DSM10027), *V. vulnificus* (DSM 10143), *V. fluvialis* (DSM 19283), *V. mimicus* (DSM 19130) and *V. alginolyticus* (DSM19130) and one locally isolated *V. cholerae*. *E. coli ATCC 35150* was employed as a negative control for all the PCR assays.

### 2.7. Statistical Analysis

The entire sampling regime was grouped into four seasons: Winter (June, July, August), Spring (September, October, November), Summer (December, January, February) and Autumn (March, April, May). We hypothesized that there is no significant difference

in mean densities of *Vibrio* spp. and physicochemical parameters across seasons. Data generated (density and physiochemical parameters) were subjected to Shapiro–Wilk normality test. Afterwards, the seasonal mean densities and values for physicochemical parameters were compared across seasons using one-way analysis of variance (ANOVA) and Fisher's least significant difference (LSD) post hoc test at $p < 0.05$. Pearson correlation analysis was performed to understand the relationship between *Vibrio* species density and physicochemical parameters at each of the WWTPs. The statistical methods were so chosen because the data we generated for each of our variables were normally distributed. Statistical Package for Social Sciences (SPSS) version 20 was used for the statically analysis.

## 3. Results

All physicochemical parameters (electrical conductivity, total dissolved solids, salinity, temperature, dissolved oxygen, biological oxygen demand, total suspended solids, turbidity and residual chlorine) measured except residual chlorine correlated positively (Table 3) with Vibrio species density at high significance level ($p < 0.01$) at ALEFF. The distribution of Vibrio species density correlated positively and significantly with temperature at TSUP ($r = 0.545$, $p < 0.01$); TSEFF ($r = 0.526$, $p < 0.01$); TSDW ($r = 0.517$, $p < 0.01$); ALUP ($r = 0.509$, $p < 0.01$) and ALDW ($r = 0.607$, $p < 0.01$), dissolved oxygen at TSEFF ($r = 0.421$, $p < 0.05$), total suspended solid at TSUP ($0.387$, $p < 0.05$), turbidity at TSUP ($r = 0.386$ $p < 0.05$), electrical conductivity at ALUP ($r = 0.449$, $p < 0.05$) and ALDW ($r = 0.331$, $p < 0.05$), total dissolved solid at ALUP ($r = 0.543$, $p < 0.01$) and ALDW ($r = 0.350$, $p < 0.05$), pH at ALUP ($r = 0.377$, $p < 0.05$) and ALDW ($r = 0.607$, $p < 0.05$), salinity at ALUP ($r = 0.390$, $p < 0.05$), residual chlorine at TSDW ($r = 0.463$, $p < 0.05$), but Vibrio species density correlated negatively and significantly with dissolved oxygen at TSUP. Temperature is the only parameter that correlated positively at high significant level ($p < 0.01$) with the distribution of *Vibrio* species density at all sampling points (Table 3).

**Table 3.** The correlation matrix between the *Vibrio* species density and water quality indexes of AL and TS WWTPs.

| | | **TS WWTPs** | | | | |
|---|---|---|---|---|---|---|
| **Site** | **Physicochemical parameters** | **Cfu/ml Corel coeff** | **Site** | **Cfu/ml Corel coeff** | **Site** | **Cfu/ml Corel coeff** |
| **TSUP** | pH | −0.327 | **TSEFF** | 0.218 | **TSDW** | 0.27 |
| | Cond | −0.076 | | −0.232 | | 0.026 |
| | TDS | −0.076 | | −0.182 | | 0.044 |
| | Sal | −0.064 | | −0.175 | | 0.014 |
| | Temp | 0.545 ** | | 0.526** | | 0.517 ** |
| | DO | −0.393 * | | 0.421* | | −0.012 |
| | BOD | 0.304 | | 0.332 | | −0.029 |
| | TSS | 0.387 * | | 0.298 | | 0.13 |
| | Turb | 0.386 * | | 0.275 | | 0.134 |
| | FreeCl$_2$ | | | 0.238 | | 0.463 ** |
| | | **AL WWTPs** | | | | |
| **Sites** | **Physicochemical parameters** | **Cfu/ml Corel coeff** | **Site** | **Cfu/ml Corel coeff** | **Site** | **Cfu/ml Corel coeff** |
| **ALUP** | pH | 0.377 * | **ALEFF** | 0.497** | **ALDW** | 0.607** |
| | Cond | 0.449 ** | | 0.488** | | 0.331* |
| | TDS | 0.543 ** | | 0.452** | | 0.350* |
| | Sal | 0.390 * | | 0.459** | | 0.246 |
| | Temp | 0.509 ** | | 0.615** | | 0.559** |
| | DO | −0.04 | | 0.615** | | 0.07 |
| | BOD | 0.212 | | 0.575** | | 0.281 |
| | TSS | −0.185 | | 0.511** | | 0.103 |

| | Turb | −0.087 | 0.630** | 0.197 |
| | FreeCl$_2$ | | 0.137 | 0.107 |

**Note:** Corel coeff, correlation coefficient; Cond, electrical conductivity; TDS, total dissolved solids; Sal, salinity; Temp, temperature; DO, dissolved oxygen; BOD, biological oxygen demand; TSS, total suspended solids; Turb, turbidity; FreeCl$_2$, residual chlorine. Significant correlation coefficient at: *p < 0.05; **p < 0.01.

The seasonal variation in the number of culturable *Vibrio* populations recovered from water samples collected from TS and AL WWTPs and the neighbouring freshwater bodies across different seasons are shown in Figures 1a and 1b. Additionally, Tables 4a and 4b show the significant echelons (P values) of the statistical comparison of the average annual and seasonal mean of presumptive *Vibrio* species observed for both AL and TS WWTPs, respectively. The means of presumptive *Vibrio* species density in both WWTPs and the receiving water bodies throughout the sampling regime ranged from 1.05 ± 0.23 to 2.08 ± 0.07 Log cfu/ml (Figures 1a and 1b). The annual mean *Vibrio* species densities at the final effluent discharge points are significantly higher than the observed densities at both the up-stream and the down-stream of the two WWTPs. Additionally, the annual mean density at the down-stream is significantly higher than that at upstream for both WWTPs. In the case of AL treatment plant, mean *Vibrio* populations (0.81 ± 0.10 Log CFU/ml), estimated from TCBS counts, were lowest during Winter (June) at the upstream point (ALUP), while the highest abundance (2.14 ± 0.00 Log CFU/ml) was observed during Winter (July) at the discharged point of effluents from the treatment plant (ALEFF). At TS WWTP, the lowest density of 0.78 ± 0.09 Log CFU/ml was observed in Winter (June) at TSUP, while the highest density of 2.07 ± 0.00 Log CFU/ml was observed in Spring (October) at TSEFF. On the other hand, density varies across seasons at the two WWTPs. The average density in the Summer was significantly higher than that for other seasons except in Autumn. Additionally, density recorded in Spring was significantly higher than that observed in Autumn and Winter, while the density observed for Autumn was significantly higher than density observed for Winter. All recovered presumptive *Vibrio* isolates were further confirmed via PCR assay, with expected amplicon sizes (~663 bp) of *Vibrio* genus-specific 16S rRNA region obtained (Figure 2). Figures 3 and 4 showed the electrophoresis gel images for duplex PCR amplicons of *V. mimicus* and *V. cholerae* and triplex PCR for the identification of *V. fluvialis*, *V. alginolyticus* and *V. vulnificus*, respectively. Additionally, the agarose gel image of conventional PCR amplicons of *V. parahaemolyticus* is presented in Figure 5. Consequently, the distribution of confirmed *Vibrio* isolates considered across all seasons is presented in Figures 6a and 6b. The figures show that a major percentage of the *Vibrio* species were recovered from TS and AL WWTP samples during warm periods of our sampling regime. The overall occurrence of *Vibrio* species was 60% (76/127) and 75% (86/115) in TS and AL WWTPs water samples, respectively. Among the positive *Vibrio* species recovered from TS WWTP (n = 127), *V. cholerae* showed the highest rate of detection (33.8%), trailed by *V. mimicus* and *V. fluvialis* (11 and 9.4%, correspondingly). The remaining confirmed species (*V. alginolyticus*, *V. vulnificus* and *V. parahaemolyticus*) comprised between 0 and 5.5%. On the other hand, the abundances of the selected confirmed *Vibrio* species (n = 115) recovered from AL WWTP water samples were 34, 19, 13, 6, 2 and 1% for *V. cholerae*, *V. mimicus*, *V. fluvialis*, *V. parahaemolyticus*, *V. vulnificus* and *V. alginolyticus*, respectively. In addition, the *Vibrio* populations from treated effluents in this study were dominated by *V. cholerae* and *V. mimicus* across all seasons (Summer, Winter, Autumn and Spring), with *Vibrio alginoliticus* and *Vibrio vulnificus* having the lowest population throughout the sampling regime. The average seasonal comparison of six confirmed clinically important *Vibrio* species from TS and AL WWTPs, respectively, is presented in Table 5. The abundances of the selected *Vibrio* species within Autumn vs. Spring, Winter vs. Spring (TS WWTP) and Winter vs. Spring (AL WWTP) were not significantly different for both treatment plants (Table 5). However, significant differences (p < 0.05) were observed when the seasonal occurrences of *Vibrio* spp. in samples from both TS and

AL WTTPs were compared for seasons: Summer vs. Autumn, Summer vs. Winter, Summer vs. Spring, Autumn vs. Winter and Autumn vs. Spring.

**Table 4.** Statistical comparison of the average annual presumptive *Vibrio* densities in TS and AL wastewater treatment plants.

| TS wastewater treatment plant | | AL wastewater treatment plant | |
|---|---|---|---|
| Site Types | p values | Site Types | p values |
| TSUP vs. TSEFF | <0.0001 * | ALUP vs. ALEFF | <0.0001 * |
| TSUP vs. TSDW | 0.007 * | ALUP vs. ALDW | <0.0001 * |
| TSEFF vs. TSDW | <0.0001 * | ALEFF vs. ALDW | <0.0001 * |

* = Significant different.

**Table 5.** Significant levels of the statistical comparison of average seasonal presumptive *Vibrio* densities in TS and AL wastewater treatment plants.

| TS wastewater treatment plant | | AL wastewater treatment plant | |
|---|---|---|---|
| Seasons | p values | Seasons | p values |
| Summ vs. Autu | 0.524 | Summ vs. Autu | 0.394 |
| Summ vs. Wint | <0.0001 * | Summ vs. Wint | <0.0001 * |
| Summ vs. Spri | <0.0001 * | Summ vs. Spri | 0.001 * |
| Autu vs. Wint | <0.0001 * | Autu vs. Wint | <0.0001 * |
| Autu vs. Spri | <0.0001 * | Autu vs. Spri | 0.009 * |
| Wint vs. Spri | 0.011 * | Wint vs. Spri | 0.014 * |

*, significantly different; Summ, Summer; Autu, Autumn; Wint, Winter; Spri, Spring.

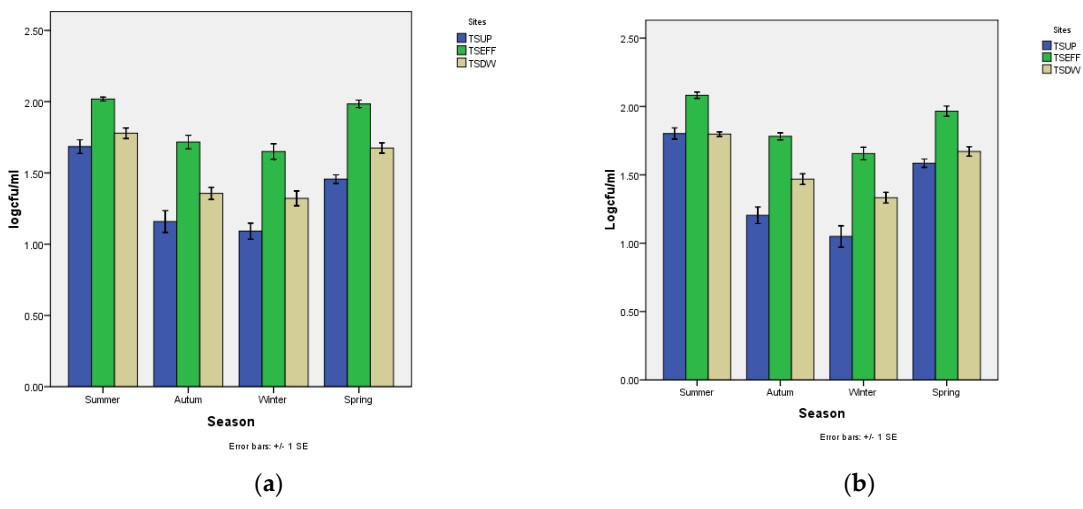

**Figure 1.** (**a**). Mean seasonal presumptive *Vibrio* species densities of TS WWTP (**b**). Mean seasonal presumptive *Vibrio* species densities of AL WWTP. TSUP, upstream discharge point; TSEFF, effluent; TSDW, downstream discharge point, bars on column showed standard error of three replicates.

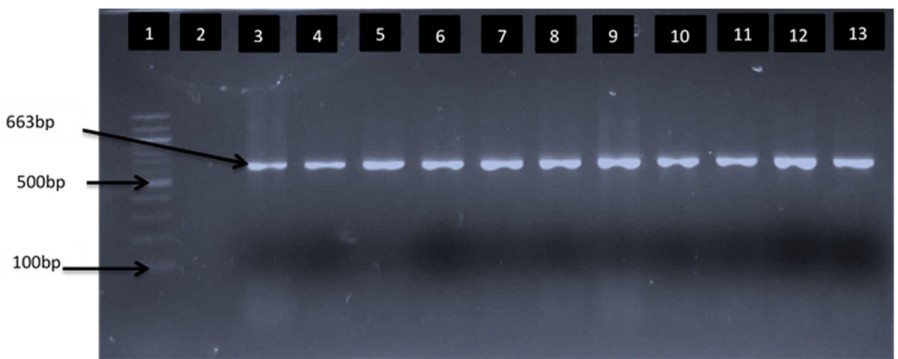

**Figure 2.** Agarose gel image showing PCR amplicons of the *fla E* gene, specific for *V. parahaemolyticus*. Lane 1, molecular weight marker (100 bp); lane 2, negative control; lane 3, positive control (DSM 19130); lanes 4–12, positive isolates.

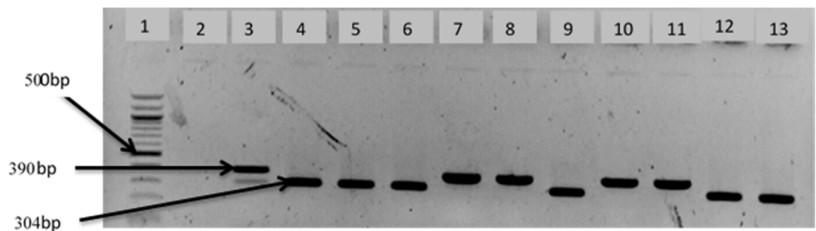

**Figure 3.** Agarose gel with duplex PCR amplicons showing *OmpW* and *vhm* gene regions specific for *V. cholerae* and *V. mimicus,* respectively. Lane 1, molecular weight marker (100 bp); lane 2, negative control; lane 3, positive control locally sourced *V. cholerae* and DSM 19130); lanes 4–6, 9, 12 and 13, *V. cholerae* positive isolates; lanes 7, 8, 10 and 11, *V. mimicus*-positive isolates.

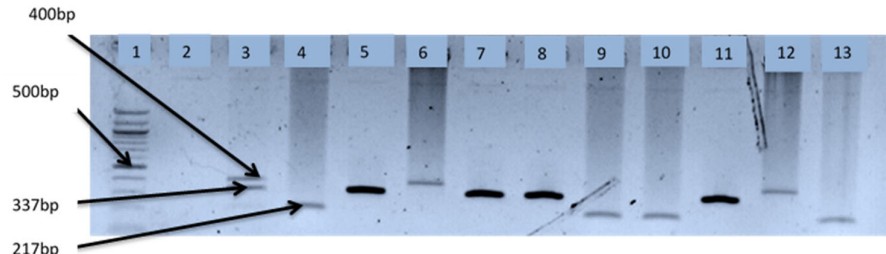

**Figure 4.** Agarose gel with triplex PCR amplicons showing *GroEl, ToxR* and *GyrB* genes regions specific for *V. vulnificus, V. fluvialis* and *V. alginolyticus,* respectively. Lane 1, molecular weight marker (100 bp); lane 2, negative control; lane 3, positive control DSM 19130, DSM 10143, DSM 19283); lanes 4, 9, 10 and 13, *V. fluvialis* positive isolates; lanes 5, 7, 8 and 11, *V. alginolyticus* positive isolates; lanes 6 and 12, *V. vulnificus* positive isolates.

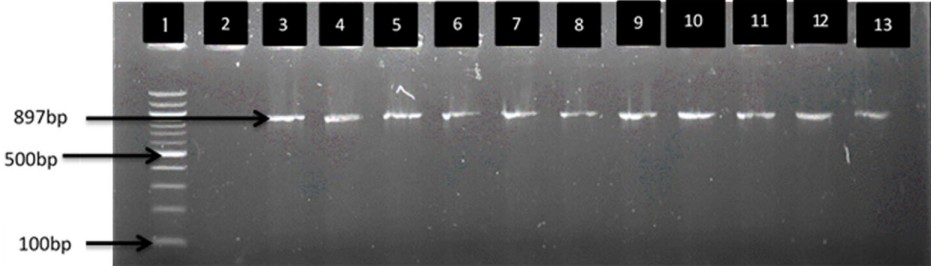

**Figure 5.** Agarose gel with PCR amplicons showing *fla E* gene specific for *V. parahaemolyticus*. Lane 1, molecular weight marker (100 bp); lane 2, negative control; lane 3, positive control (DSM 19130); lanes 4–12, positive isolates.

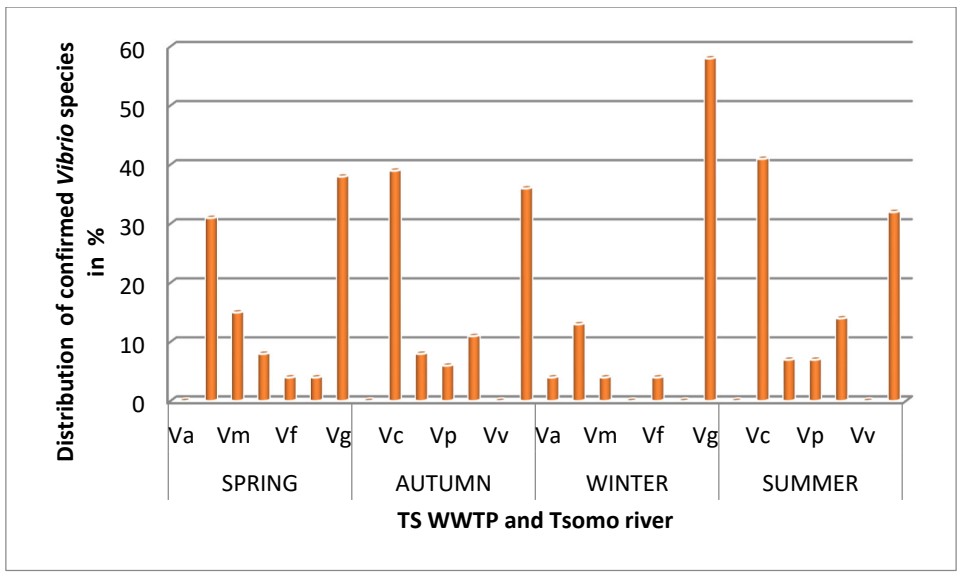

(**a**)

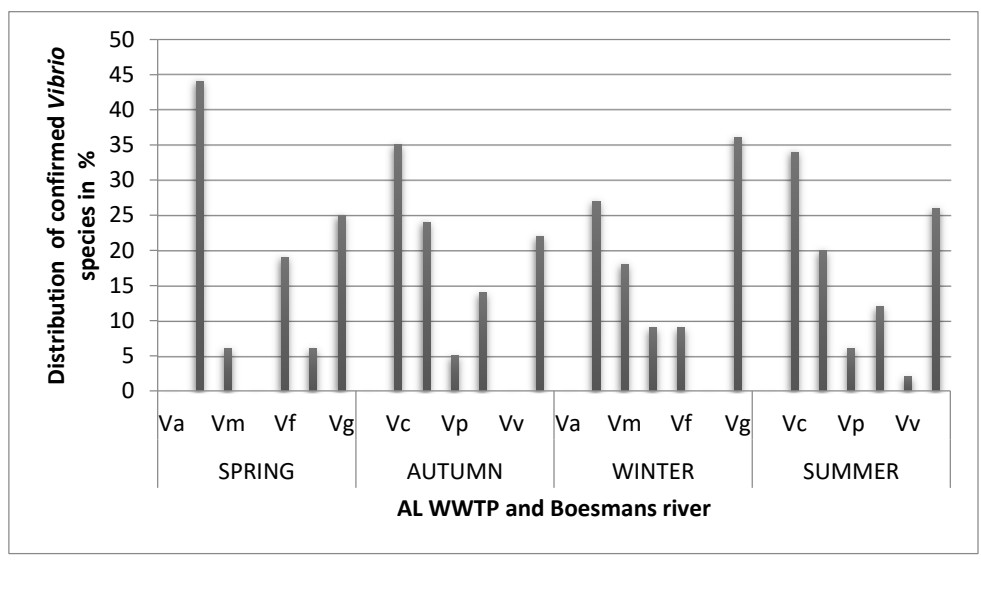

(**b**)

**Figure 6. (a**): Percentage distribution of *Vibrio* spp. of interest admits isolates recovered from TS WWTP and receiving water bodies (Tsomo river) for each season. Vc, *V. cholerae*; Vm, *V. mimicus*; Vp, *V. parahaemolyticus*; Vf, *V. fluvialis*; Vv, *V. vulnificus*; Vg, *Vibrio* genus; (**b**): Percentage distribution of *Vibrio* spp. of interest among the isolates recovered from the AL WWTP and receiving water bodies (Boesman river) for each season. Va, *V. alginolyticus*; Vc, *V. cholerae*; Vm, *V. mimicus*; Vp, *V. parahaemolyticus*; Vf, *V. fluvialis*; Vv, *V. vulnificus*; Vg, *Vibrio* genus.

## 4. Discussions

In many countries, especially in the developing world, freshwater resources are severely contaminated with pathogens leading to various waterborne disease outbreaks [32]. *Vibrio* species occur naturally typically surviving in freshwater and marine environments. Their existence and role in human infections and *Vibrio*-facilitated biotransformation and remineralisation processes have been well documented [33–36]. In addition, the discharge of inadequately treated effluent in surface water often result in fluctuation of microbial community, with consequences to water quality and water-borne pathogen

contamination [37]. Recently, [38] reported that microbial community structure of freshwater resources are largely shaped by several key physicochemical variables. These environmental parameters are important factors in ascertaining the transmission of pathogens and environmental persistence (disease ecology). They are usually employed in projecting and preventing cases of infectious disease outbreaks arising from waterborne pathogens [39,40].

On the other hand, the receiving water bodies higher bacterial density may cause rapid depletion of DO due to the higher rate of decomposition of organic matter, which are more abundant through discharged effluent from WWTP [41]. This explains why a negative correlation between DO and *Vibrio* density in surface waters nearby wastewater discharge point was observed in this study. The microorganisms' correlation with TSS and TDS show them as agents of major drivers of either spatiotemporal distribution of the bacteria in aquatic milieu or as factors needed to sustain their physiological status within the environment. The strong correlations between conductivity and TDS, conductivity and salinity and TDS and salinity could be ascribed to their reliant on ionic effluence loadings of freshwaters resources. The findings from this study are slightly in agreement with [42], where turbidity positively and significantly stimulate bacterial load.

Temperature is regarded as one of the most important factors affecting microbial growth and survival in the environment [43,15]. Positive correlation exhibited by temperature indicates its importance on distribution and abundance of *Vibrio* spp. in freshwater. Most recently, several studies have reported a varying range in temperature requirements for *Vibrio* species growth. Their findings further revealed that these pathogens grow extremely well at a mesophilic temperature range of 15 to 45 °C for most strains [44,45]. The significant difference observed in the presumptive seasonal density of *Vibrio* species portrays difference in season and sampling sites with respect to the month of sampling. The low presumptive densities during Winter (June–August in South Africa) are associated with low temperatures, suggesting that *Vibrio* species might have reverted to the viable but non-culturable (VBNC) state during Winter, when environmental factors become unfavourable [44]. Outbreaks of *Vibrio* infections usually occur in the warm months of the year [46]; hence, public health officers should be more vigilant for *Vibrio* infections outbreaks during this period.

Most South African wastewater treatment works disinfect wastewater by chlorination prior to discharge into receiving watersheds [47]. The goal is to remove pathogens from wastewater. Moreover, chlorine can be harmful to aquatic life if high concentrations are discharged into the environment [48]. It is, therefore, regulated before being discharged into the environment as free or residual chlorine in South Africa [49]. To achieve this goal, residual chlorine is maintained at sufficient levels and in contact with the microbial community in the chlorination tank. The residual chlorine concentration in this study ranged between 0.00 and 1.41 mg/l (Table S1 & S2). The chlorine residual concentration fell below the minimum recommended concentration in TS WWTP and exceeded the maximum limit of 0.6 mg/l in AL WWTP of the year under review, except during the autumn season and July 2017. Similar ranges have been reported for chlorine residual concentration in South African wastewater treatment plants [50] and indicate that some South African waterworks do not comply with stipulated standards with reference to free chlorine residual concentration. A positive correlation was observed between chlorine residual concentration and *Vibrio* density (Table 6); the abundance of *Vibrio* species shows they can withstand the presence of residual chlorine in the treated effluents. The high residual chlorine concentration detected confirmed that the dosed chlorine was more than sufficient and possibly indicated the presence of chlorine-resistant strains of *Vibrio* species.

**Table 6.** P values showing significant differences in the average seasonal PCR confirmed *Vibrio* species in TS and AL wastewater treatment plants.

| TS wastewater treatment plant | | AL wastewater treatment plant | |
|---|---|---|---|
| Seasons | p values | Seasons | p values |
| Summ vs. Autu | 0.108 * | Summ vs. Autu | 0.055 * |
| Summ vs. Wint | 0.001 * | Summ vs. Wint | <0.0001 * |
| Summ vs. Spri | 0.003 * | Summ vs. Spri | <0.0001 * |
| Autu vs. Wint | 0.013 * | Autu vs. Wint | 0.002 * |
| Autu vs. Spri | 0.37 | Autu vs. Spri | 0.006 * |
| Wint vs. Spri | 0.516 | Wint vs. Spri | 0.376 |

*, significantly different; Summ, Summer; Autu, Autumn; Wint, Winter; Spri, Spring.

However, faulty clarifiers, sludge reticulation due to aerator breakdown and stabilization ponds that are not associated with disinfection have been identified during sampling regime with high *Vibrio* densities. These observations were consistent with the findings of [47] that reported polluted effluents in their study.

Our findings show that successful isolation and identification of varying *Vibrio* species via membrane filtration and molecular (PCR) methods reveal the large-scale presence of waterborne pathogens in treated wastewater effluents, reaffirming their ability to thrive and survive conventional wastewater treatment processes, as previously reported [51,52]. *Vibrio* species were recovered from all samples assessed throughout the sampling regime of this study. In line with our observation, the occurrence and prevalence of *Vibrio* species in wastewater have been documented by other researchers [53,54,14]. Our finding implies that there exists risk of infection with the potential pathogenic *Vibrio* species from effluents and their receiving water bodies. The infectious dose of *V cholerae* required to cause clinical disease varies by the mode of administration; with water $10^3$–$10^6$ organisms and food, fewer organisms ($10^2$–$10^4$) are required to produce disease. *Vibrio* species are capable of causing gastroenteritis, septicaemia and wound infection and may be deadly for persons in immune-compromised state, though the resulting infection is frequently self-limited [55].

[56] also recorded the abundance of *Vibrio* species in treated effluents during the Summer season. The detection of six notable *Vibrio* pathogens (*V. cholerae, V. mimicus, V, alginolyticus, V. fluvialis, V. vulnificus* and *V. parahaemolyticus*) from TS and AL WWTPs is consistent with the observation of [57] that the abundance of *Vibrio* species varied widely in environmental waters. This findings further strengthens existing calls for the inclusion of emerging bacterial pathogens, including *Vibrio* species, as water quality indicators by the South African Department of Water Affairs.

Our observation on both wastewater treatment revealed it is a high risk to plants with potential danger to the environment, which calls for urgent attention. The recent Green Drop report (2014), which serves as a means of regulating WWTPs in South Africa, categorized TS and AL WWTPs as critical and high-risk WWTPs, respectively, further supports our observations [58,59]. The Green Drop of 2014 identified some of the challenges facing the AL and TS WWTPs, which includes effluent non-compliance, failure of wastewater treatment plant response management and operating capacity that exceeds design capacity. This finding indicates that treated effluents could serve as maintenance reservoirs for dominant *Vibrio* species (*V. cholerae* and *Vibrio mimicus)* amidst the confirmed *Vibrio* pathogens. The treated effluents could also serve as transient reservoirs for *Vibrio alginoliticus* and *Vibrio vulnificus.* These results correspond well with the data obtained from the study in the Eastern Cape Province, South Africa and United Kingdom [60,61]. The direct isolation and confirmation of *Vibrio* pathogens from TS and AL WWTPs further revealed the influence of seasonal variations such as temperature that affect the distribution of *Vibrio* pathogens in aquatic milieu [62,38].

Consequently, the persistence of these pathogens in treated effluents could be attributed to the impact of water run-offs carrying intestinal waste from livestock in neighbouring public slaughterhouses, birds and healthy carriers among the local human population [63], making the sewage conducive for the survival and proliferation of enteric pathogens, such as *Vibrio* species [14]. The recovery of notable non-halophilic (*V. cholerae* and *V. mimicus*) and halophilic (*V, alginolyticus, V. fluvialis, V. vulnificus* and *V. parahaemolyticus*) *Vibrio* species in the same niche in this study is significant for the following reasons. First, the risk of acquiring infections associated with these pathogens might be on the rise, considering the concentration and diversity of *Vibrio* pathogens found within the same niche. Secondly, the coexistence of the various *Vibrio* species might result in interspecies interaction and the potential exchange of virulent gene markers through horizontal gene transfers [64]. Some of the known traits of *Vibrio* pathogens can be linked with quorum sensing, since these bacteria have been reported to be able to express their virulence factors through signalling molecules [65]. The surge in cases of infection could be partially attributed to global warming, as several, sporadic reports of *V. alginolyticus, Vibrio vulnificus* and *Vibrio parahaemolyticus* infections have also been recognized in regions with temperate climates, e.g., the United States of America [66] and, more recently, in Europe [15].

## 5. Conclusions

The results obtained via conventional bacteriology, with the aid of membrane filtration technique and PCR, show the treated effluents and receiving water bodies as environmental reservoirs of *Vibrio* pathogens considered in this study. Perpetually changing environmental conditions such as increasing surface water temperatures can significantly influence the risk of infections related to potentially human-pathogenic *Vibrio* species. Such changes may also affect temperate regions with mild subtropical climates such as the South Africa. Detection of high residual chlorine concentrations were observed; thus, this calls for concern on the issue of chlorine dosing because higher residual concentrations that are not compliant to both South Africa and general standards are being discharged into the environment, which is not good for aquatic life. Therefore, augmenting the inefficient chlorine disinfection process at the current plant with maturation ponds, ozonation or ultraviolet radiation could undoubtedly aid the treatment plants in discharging treated effluents with zero *Vibrio* species. Considering, the public health implication in the use of inadequately treated effluents, we advocate for regular monitoring of water reservoirs for possible microbial pathogens to allow for early response by public health authorities (e.g., prevention and treatment measures to combat relevant diseases). We further recommend regular monitoring of water reservoirs (treated effluents and receiving water bodies) to ensure early control of potential outbreaks of vibriosis and cholera.

**Supplementary Materials:** The following are available online at www.mdpi.com/2073-4441/13/7/932/s1, Table S1: Tsomo WWTP microbial load and physicochemical, Table S2: Alicedale WWTP microbial load and physicochemical.

**Author Contributions:** A.I.O., conceptualization, provided materials for the study, proof-read the manuscript; O.C.A., conceptualization, structured the methods used in the study, carried out the experiments, wrote the original draft; O.A.E., optimized the methods and analysed the data.

**Acknowledgement:** We thank the South Africa Medical Research Council and the South Africa Water Research Commission for financial support. We also thank Dr. Fiyin Olaitan for her assistance in editing the whole manuscript.

**Institutional Review Board Statement:** Not applicable.

**Informed Consent Statement:** Not applicable.

**Data Availability Statement:** All data generated or analysed during this study are included in this manuscript.

**Ethics Statement:** None required.

**Conflicts of Interest:** We have no conflicts of interest to disclose.

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
