# Peer review of "Distribution and Public Health Significance of Vibrio Pathogens Recovered from Selected Treated Effluents in the Eastern Cape Province, South Africa"

_water, doi:10.3390/w13070932_

Round 1

Reviewer 1 Report

The current study is important in terms of public health. However, there are some major and minor drawbacks which are listed below and in the attached manuscript.

1- Give a novelty statement at the end of the introduction. 2- No hypothesis is provided. Give a clear-cut hypothesis. 3- No information about means and replicates is provided in the figure captions. 4- Correct the symbol of probability throughout the manuscript either use P or p. 5- No conclusion is provided which is a major drawback of this manuscript.   For minor changes please see the attached file.

Author Response

1- Give a novelty statement at the end of the introduction

Response: Interestingly, this is the first time that this kind of study will be carried out at the selected WWTPs.

2- No hypothesis is provided. Give a clear-cut hypothesis.

Response: We hypothesized that there is no significant difference in mean densities of Vibrio spp. and physicochemical parameters across seasons. Data generated (density and physiochemical parameters) were subjected to Shapiro-Wilk normality test. 

  1. No information about means and replicates is provided in the figure captions.

Response: It has been addressed in the attached manuscript

  1. Correct the symbol of probability throughout the manuscript either use P or p. 5- 

Response: It has also been effected in the attached manuscript

  1. No conclusion is provided which is a major drawback of this manuscript.

Response: The conclusion section has been included

Reviewer 2 Report

Thank you for the opportunity to revise this manuscript, as it deals with a very important public health issue (i.e., water quality and microbial contamination). I found the study design appropriate, and the discussion is supported by the findings. Overall, I believe the manuscript is well presented and it could be of interest to the readers. I have only a few minor suggestions, reported below. In particular, the abstract section is too long, and it could be shortened by deleting the first introductive sentences. Secondly, the Introduction section reports sentences that are more appropriate for the discussion section (see lines 70-78). I suggest moving or deleting them.

Author Response

Thanks for your observation but we are of the opinion that the introductory sentences gives insight into the concept of the manuscript

Also the introduction section between line 70-78 serves as motivation

Reviewer 3 Report

The manuscript entitled "Distribution and Public Health Significance of Vibrio Pathogens Recovered from Selected Treated Effluents in the Eastern Cape Province, South Africa", aimed to evaluate treated effluents and watersheds in South Africa, as potential reservoirs of Vibrio pathogens.

The study falls within the aims and scope of the Journal, and the methods are adequately described.

The discussion section is thorough, and the study's results are clearly presented.

I have some minor comments. In particular, it is not clear if the protocol declared in sub-section 2.5 "Confirmation of probable Vibrio isolates" is referred to reference no. 31 (authors' laboratory protocols), or it is linked to a different (not declared) PCR assay protocol. I suggest clarifying this aspect.

In the "2.7 Statistical analysis" section, the authors stated their data were normally distributed. How did they assess the normality of distribution? I suggest the authors add the description of the statistical approach used at the beginning of this sub-section.

Finally, there are several typos in the text. I suggest a revision.

Author Response

I have some minor comments. In particular, it is not clear if the protocol declared in sub-section 2.5 "Confirmation of probable Vibrio isolates" is referred to reference no. 31 (authors' laboratory protocols), or it is linked to a different (not declared) PCR assay protocol. I suggest clarifying this aspect.

Response:

This aspect of the manuscript has been clarified ‘as described in one of the earlier study from our laboratory’

In the "2.7 Statistical analysis" section, the authors stated their data were normally distributed. How did they assess the normality of distribution? I suggest the authors add the description of the statistical approach used at the beginning of this sub-section.

Response:

Data generated (density and physiochemical parameters) were subjected to Shapiro-Wilk normality test.
